# The Current State of Knowledge of Shea Butter Tree (*Vitellaria paradoxa* C.F.Gaertner.) for Nutritional Value and Tree Improvement in West and Central Africa

Patrick Bustrel Choungo Nguekeng [1,2,*], Prasad Hendre [3], Zacharie Tchoundjeu [2], Marie Kalousová [1], Armelle Verdiane Tchanou Tchapda [1,2], Dennis Kyereh [1], Eliot Masters [4] and Bohdan Lojka [1,*]

1   Department of Crop Sciences and Agroforestry, Faculty of Tropical AgriSciences, Czech University of Life Sciences Prague, Kamýcká 129, Suchdol, 165 00 Prague, Czech Republic; kalousovam@ftz.czu.cz (M.K.); Armelle_verdiane@ftz.czu.cz (A.V.T.T.); kyereh@ftz.czu.cz (D.K.)
2   Higher Institute of Environmental Sciences (HIES), Yaoundé 35460, Cameroon; Z.Tchoundjeu@gmail.com
3   World Agroforestry (CIFOR-ICRAF), United Nations Avenue, Gigiri, P.O. Box 30677, Nairobi 00100, Kenya; P.Hendre@cgiar.org
4   Nelson Marlborough Institute of Technology, Nelson 7010, New Zealand; Eliot.Masters@nmit.ac.nz
*   Correspondence: Patrick_bustrel@ftz.czu.cz (P.B.C.N.); lojka@ftz.czu.cz (B.L.);
    Tel.: +42-(0)776-187-392 (P.B.C.N.); +42-(0)734-170-763 (B.L.)

**Abstract:** *Vitellaria paradoxa* (C.F.Gaertn.) is a multi-purpose tree species distributed in a narrow band across sub-Saharan Africa. The species is integrated into cropping and agroforestry systems as a nutritional and economic resource, which provides a range of environmental services. Integration of the species into land-use systems provides an essential source of livelihoods and income for local populations. The economic potential of the shea butter tree derives from its edible products, which also serve cosmetic and pharmaceutical applications. To understand the current state of knowledge about *V. paradoxa*, this paper summarizes information about the ecology, population structure, and genetic diversity of the species, also considering compositional variation in the pulp and kernels, management practices, and efforts towards its domestication. Despite the great potential of the shea butter tree, there are some gaps in the understanding of the genetics of the species. This review presents up-to-date information related to the species for further domestication and breeding purposes.

**Keywords:** agroforestry; biodiversity conservation; domestication; genetic resources; livelihoods; molecular marker; sub-Saharan Africa

## 1. Introduction

Non-timber forest products (NTFPs) contribute to the livelihood improvement of local communities [1]. They play a significant role in poverty alleviation among rural farmers by providing food security and reducing malnutrition [2,3]. Today, many crops and trees on our planet are utilized for human consumption, providing nutritional and economic resources in smallholder farming systems [4]. Many of these edible plants are indigenous local fruit tree species adapted to their ecosystem, with potential for domestication. Local communities face difficulties due to overexploitation of forest and woodland resources, which are rapidly depleting [5–7]. With growing populations, and demand for food expected to double over the next 30 years, solutions are required to address future food and nutritional security in smallholder agricultural systems, including aspects of nutrition, plant science, genomics, and agroforestry [8,9].

Trees play a significant role in African agricultural landscapes and agroforestry parklands [10], contributing substantially to rural livelihoods [11]. Tree genetic resources can improve soil fertility, among other environmental services, and provide sources of livelihood through products such as timber, fruits, and medicines [12]. Other ecosystem

services provided by trees include conservation of associated biodiversity, by providing food and shelter for living organisms, regulation of temperature, shade, and carbon sequestration [13].

The African Orphan Crop Consortium (AOCC) has been working with World Agroforestry (the International Centre for Research in Agroforestry, ICRAF) on genomic resources for the improvement of underutilized species which have hitherto received little attention from the scientific community, with a particular focus on trees. The objective of this consortium is to modernize and improve the efficiency of plant breeding practices to improve yield, climate resilience, and nutrition thus improving livelihoods and quality of life of small holder farmers [14–17]. Target species provide vitamins, minerals, and essential micronutrients in the diets of producer communities. Early results on two *Artocarpus* species, jackfruit (*Artocarpus heterophyllus*) and breadfruit (*Artocarpus altilis*), have resulted in their genome sequencing and publication [18]. Some genomic data have also been generated for white Acacia (*Faidherbia albida*), marula (*Sclerocarya birrea*), and moringa (*Moringa oleifera*) [19]. These species are classified as crucial African orphan crops with strong potential to improve the livelihoods of farmers [20].

One such species, the shea (butter) tree (*Vitellaria paradoxa* C.F.Gaertn.), known as *karité* in French, is an indigenous fruit tree belonging to the Sapotaceae family. There are two primary sub-species, namely *nilotica* and *paradoxa*. It is found in a belt of Sudanic vegetation extending south of the Sahel within the western, central, and eastern regions of sub-Saharan Africa (SSA) [21–23]. The shea tree is an essential source of income and livelihood, generating ecological and environmental benefits in the traditional parkland agroforestry system [24]. It is a multipurpose tree, yielding nutritious fruit pulp and kernels and a range of other derived products with edible and medicinal applications. The kernel lipids, known as shea butter, are consumed by rural households, and sold on local markets across the zone as an edible fat of great importance [24]. It is also used as an industrial feedstock in global supply chains serving the confectionery, cosmetic, and pharmaceutical industries. Aside from these value-added applications, the wood of the shea tree is used to produce charcoal of very high quality in terms of its density and performance [25]. Shea butter is referred to as 'women's gold' due to the livelihood benefits that women farmer derive from shea production and processing across the value chain [26]. It is one of the priority species in the larger tree domestication and improvement program implemented on a long-term basis by World Agroforestry.

The AOCC has prioritized the shea tree as one of the 101 local plant species identified as crucial to develop livelihood options for many families across SSA [16]. However, the species has a weak regeneration ability in some areas (particularly in Cameroon) and is facing survival challenges due to deteriorating ecological conditions, poor soil, poor pollination, parasites, and anthropic pressures [25]. Domestication using traditional and advanced technologies, such as genomics, offers a means of selection for the development of new plant varieties that can help farmers. Scientists have been using phenotypic studies and environmental characterization in genetic variation to assess variation in compositional properties [2,27–30]. The shea tree can be considered a high-priority genetic resource of SSA countries [8,31]. The main objective of this review paper is to summarize the available information from the existing literature, identify research gaps relevant to the domestication potential of the shea tree, and consider potential for development of improved genetic material to support the needs of smallholder farmers. Specifically, the paper focuses on the genetic diversity, compositional and morphological variation, and population structure of *V. paradoxa* populations across their natural range.

## 2. Taxonomy and Botanical Description

*V. paradoxa* belongs to the Sapotaceae family, which consists of trees, lianas, woody lianas, and shrubs, divided into 53 genera [32], mostly categorized as trees in the tropical and subtropical regions of South America and Asia [33,34]. Many of the species are

characterized by sticky white latex, often found in the bark, branches, leaves, and fruits. Sapotaceae species often occur, as in slow-growing species, in dry conditions [35,36].

According to Plant Resources of Tropical Africa (PROTA), *Vitellaria* is comprised of a single species—*V. paradoxa.* Two subspecies are recognized in *V. paradoxa* subsp. *paradoxa* (synonym: *Butyrospermum parkii* (G.Don) Kotschy) and *V. paradoxa* subsp. *nilotica* (Kotschy)—A.N.Henry, Chithra, and N.C.Nair (synonym: *Butyrospermum niloticum* Kotschy). The *paradoxa* subspecies has a less dense and shorter indumentum and slightly smaller flowers than the *nilotica* subspecies. The former occurs from Senegal to the Central African Republic; the latter is found in Sudan and Uganda, with small populations in Ethiopia and the Democratic Republic of the Congo. The ranges of the two subspecies do not overlap, although they come within 175 km of each other at the divide between the drainage basins of Lake Chad and the Congo River to the west, and the Nile to the east and north-east [37]. The species are both used for the same purposes, but the *paradoxa* subspecies are most common and prioritized [38]. This species appears on the red list of threatened species to be vulnerable, according to the International Union for Conservation of Nature (IUCN) (https://www.gbif.org/species/2886750 (accessed on 7 December 2021)

*V. paradoxa* is a medium size tree, with a height of 7–25 m. Mature trees have a diameter ranging from 0.3 to 1 m. The tree possesses a thick bark, longitudinally and deeply fissured [39,40]. The leaves are in dense clusters, spirally arranged at the ends of stout twigs. The petiole length ranges from 5 to 15 cm long. Juvenile leaves are rust-red and pubescent, later coriaceous, glabrous, dark green, shiny, 12–25 cm long, and 4–7 cm wide [34]. The tree bears several flowers clustered in the axils of terminal leaves or leaf scars on leafless twigs. The number of flowers per inflorescence is highly variable and could be more than 100. The flower is hermaphroditic and actinomorphic, enveloped at the base of its peduncle by a very small bract [41,42]. The creamy white flowers, very fragrant and honey-bearing, are carried by long pedicels (22–25 cm) (Figure 1a) [43–45].

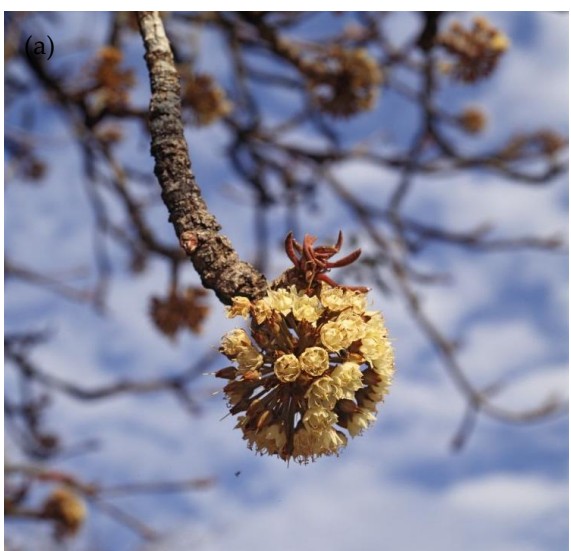
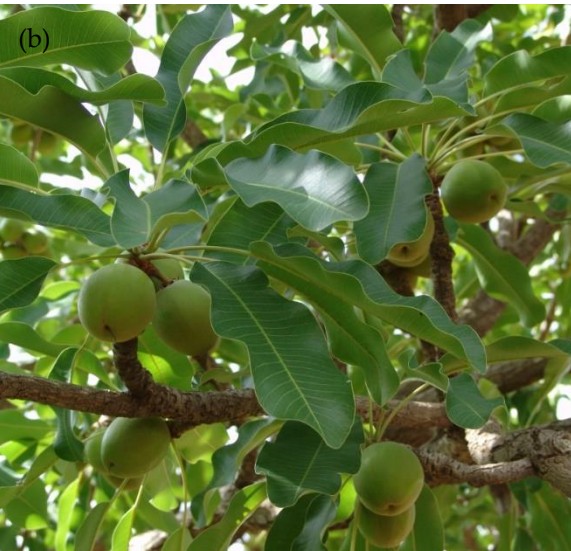

**Figure 1.** *Cont.*

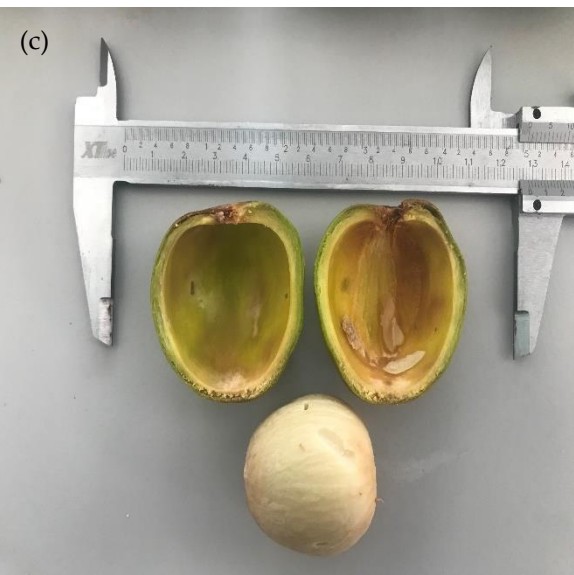
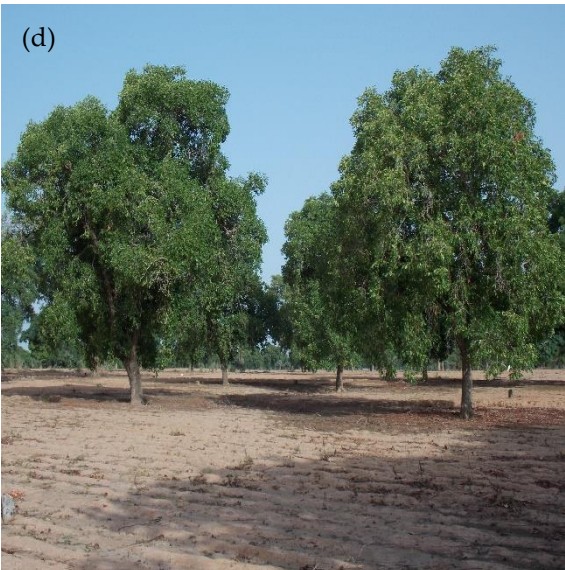

**Figure 1.** *Vitellaria paradoxa* tree. (**a**) Inflorescence of *V. paradoxa* subsp. *nilotica*, Uganda; (**b**) fruit and leaves; (**c**) open fruit from Western Cameroon provenance showing nut and flesh; (**d**) shea parkland in Mali. Photos: The Shea Project (a), E. M. (b and d) and P.B.C.N. (c).

The shea fruit can be described as a globose to ellipsoid berry, measuring 4–5 cm × 2.5–5 cm, with a weight of 20–30 g. The fruit appears to be a yellow, green, or yellow berry with thick pulp with one oval or round red-brown seed—the shea nut. A fragile, shiny shell surrounds it with a broad base with a large, round, rough hilum. During its growth, the fruit is initially green but turns yellowish-green to brown upon maturity. The seed coat is relatively thin, shiny, with a broad scar; the kernel consists of two thick, fleshy, closely addressed cotyledons without an exerted radicle [46–48]. Seedlings with hypogeal germination cotyledons remain in the seed with an epicotyl 3–4 cm long. They bears stipulated rudimentary leaves (Figure 1b,c) [37,49,50].

### 3. Ecological Requirements

*3.1. Ecology*

*V. paradoxa* is native to Gambia, Senegal, Guinea, Côte d'Ivoire, Mali, Burkina Faso, Niger, Ghana, Benin, Togo, Nigeria, Cameroon, Chad, Central African Republic, South Sudan, Uganda Democratic Republic of the Congo, and Ethiopia [39,51–56]. The species is typical of the West African savanna, but also in the southern Sahel (Figure 1d). The tree grows mainly at an altitude of 100–600 m and annual temperatures of between 25–29 °C [43]. It grows in areas with a yearly rainfall of 600–1400 mm and a 5–8-month dry season [56,57]. The tree is suited to sandy, iron-rich soils with good humus content for optimal growth and development, but it can sometimes grow in heavier soils. It prefers neutral pH, but also withstands acidic soils [39]. Shea trees require good drainage—and although somewhat adaptable to site hydrology, will not tolerate a high water table. *V. paradoxa* roots are extensive, and can tolerate extended dry seasons [34,58].

The shea tree represents 80% of the woody vegetation in agroforestry parklands in Burkina Faso [46]. Findings suggest that other valuable tree species could protect older *V. paradoxa* trees against bush fires [59]. *V. paradoxa* is dominant and sometimes occupies the same ecological niches as *Acacia Senegal* (L.) Wild., *Annona senegalensis* Pers., or *Parkia biglobosa* (Jacq.) G.Don [60]. From the data, it could be concluded that each species characterizes a specific agroforestry parkland, which could be described as a land-use system included in the very general category of multi-purpose trees on farmlands [61]. A survey was carried out to assess the number of trees available in some areas. In 1946, 230 shea butter trees/ha were reported in the Sudanian savanna zone [62]. An average of 7 trees/ha was determined in Uganda, while in Benin, five trees/ha were counted [54,63]. The decline

in shea tree population densities may be the result of firewood and charcoal production, agricultural development, and biofuel crop production, contributing to removing shea trees from the farms [64,65], but it is important to note that classification criteria (e.g., maturity) may lead to significantly different estimates.

### 3.2. Phenology

The shea butter tree is a deciduous species that loses its leaves during the dry season between November and March. Flowering occurs on leafless trees from December to April, depending on the region [21], whereas new leaves appear immediately after flowering. In Mali, flowering starts in November and ends in May. Fruiting begins in December and ends in July, and harvesting occurs from June to August, depending on the sites. It is important to note that the fruiting ability correlates with the flowering rate. A low level of flowering leads to a low level of fruiting [66]. The flowers are pollinated by stingless honeybees, identified as primary pollinators in some countries, including Ghana, Burkina Faso, and Uganda [21,34,42,67,68]. There is considerable variation in seasonal periodicity of the phenological cycle, with successive stages beginning in the northern and western ranges, and extending eastward. Fruit starts to ripen from March to August and is harvested mainly from April to September (Table 1).

**Table 1.** Phenology and periodicity of *Vitellaria paradoxa* in West and Central Africa [69–71].

| Phenology | Period |
| --- | --- |
| Falling of leaves | November–March |
| The emergence of young flower buds | November–April |
| Flowering | December–April |
| The emergence of young leaves | December–April |
| Fruit development | December–April |
| Fruit ripening | March–August |
| Fruit harvest | April–September |

### 3.3. Physiology and Reproductive Biology

Bees are the primary pollinators of shea trees and their activity of is highly affected by temperatures. The increase in temperature in the dry season could negatively affect the bees' movement in the area. Some tree species in the same landscape, such as *Mangifera indica*, *Vitex doniana*, and *Combretum* spp., which also coincide with the flowering of Shea trees, could divert the attention of bees from shea trees. The reduction in bee activity and forest/savannah burning may contribute to low fruit set in the shea populations [31]. To sum up, the reproductive success of the shea tree is very limited, and further research needs to be conducted to understand its driver well and put better strategies in place for effective fructification.

### 4. Uses

Local populations commonly use several shea trees parts (fruit, leaves, bark, and roots) for diverse purposes. The shea tree is an economically important tree species, providing NTFPs for local communities. They include various ethnobotanical uses due to a range of phytochemical compounds [72], which have notable nutritional attributes (Table 2) [73–76]. Parts of the tree are used in traditional medicine to treat various diseases and injuries, including stomach aches, headaches, fever, and jaundice [77]. With the use of bark and leaves, skin problems such as dryness, sunburn, burns, ulcers, and dermatitis could be treated [24]. The latex is used as glue when mixed with palm oil, while the bark is used in traditional medicine [34]. Finally, shea butter trees provided fodder for animals at 70% of surveyed households in Nyankpala, in the Northern Region of Ghana [25,78].

**Table 2.** Different uses of Shea butter tree adapted from [70,79].

| Type of Use | Uses | Parts Used |
|---|---|---|
| **Food** | Raw or in the form of drinks and jam | Fruit pulp |
| | Cooking oil [80] | Kernel |
| | Nectar processed into honey by bees | Flowers |
| **Industry** | Soap making [46] | Butter |
| | Cream for hair and skin, shampoo [81] | Butter |
| | Replacement of cocoa butter in chocolate | Butter |
| **Feed** | | Flowers |
| **Pharmacopoeia** | Stomach pain, headaches, eye problems [82] | Leaves, bark |
| | Throat pain, treatment of wounds, rheumatism [82] | Butter |
| | Diarrhoea, stomach problems, teeth cleaning | Root |
| | Malaria | Kernels |
| | Gynaecological problems [24] | Latex |
| | Facilitation of delivery lactation [24] | Bark |
| **Firewood** | | Trunk and branches |
| **Lighting** | | Butter |
| **Cosmetics** | Cream, perfume, shampoo [81] | Butter |
| **Handcrafts** | Musical instruments [34] | Bark, sap (used as glue/adhesive) |
| **Hunting** | The stem latex used to capture small animals [83] | Latex and glue |
| **Pesticides** | [81] | Seed residue after extraction of butter |
| **Soil conservation** | [84] | Leaves |

*4.1. Fruit Pulp and Kernels*

The fruit consists of a sweet, moist pulp surrounding an oil-rich seed. Highly perishable when ripe, the fruit is consumed fresh soon after collection, providing calories and micronutrients during the annual 'hungry season' when stored foods are in short supply, and agricultural labor is at a premium [85], the pulp can be dried for storage [86]; the pulp contains glucose, protein (0.8–4.4%), vitamins B and C, and minerals notably including calcium, iron, magnesium, and phosphorus [51,72,75]. The kernels have a rich oil called shea butter that is heavily used by local populations [64,80]. It is the essential dietary lipid within the shea parkland; it provides a characteristic taste to traditional dishes [34]. As a feedstock, the shea kernel is also traded internationally to meet industrial demands within the confectionery, cosmetic, and pharmaceutical industries [24,70,87]. Shea butter is traditionally used to treat arthritis due to the presence of compounds with anti-inflammatory bioactivity. Shea butter contains triterpene and cinnamic alcohols and esters [74,88,89].

*4.2. Roots*

In northern Nigeria, roots are used as chewing sticks to clean teeth. Moreover, they are also used in traditional medicine to treat diarrhea and stomach pain, sometimes also for dysentery and gastro-intestinal pain [77].

*4.3. Bark*

The bark is boiled in water for medicinal purposes. It is used to treat diabetes in some communities in Ghana. Furthermore, in some West African countries, including Senegal and Guinea, infusion of the bark crushed together with the bark of *Ceiba pentandra* and salt

is used to treat worm infestations in livestock [77]. In work carried out in Guinea Bissau, authors have reported that bark infusion could treat diarrhea, dysentery, gastric problems, and even leprosy [25,37].

### 4.4. Seed Husk

Husks are mainly used by local communities in northern Ghana for plastering traditional mud houses. Before use, they will pound the husk and make it into a paste. This treatment is helpful because it can help to seal the mud surface against the elements, and it also serves a decorative function.

### 4.5. Timber and Environmental Services

In parkland systems, the shea butter tree provides high-quality, termite-resistant timber [90]. The physical properties of the tree include moisture content ($19.89\% \pm 6.67\%$) and density ($1.22 \pm 0.18\,\mathrm{g/cm^3}$) [40]. The tree also plays a vital role in climate change mitigation by improving soil and water conservation in parkland agroforestry systems [33,91–93].

## 5. Traditional Processing and Products Obtained from the Shea Fruit

The shea tree is considered the second most important oil crop in Africa after the oil palm tree [94]. Following de-pulping (for consumption or conservation), the dried shea kernels are processed into shea butter through grinding, roasting, milling, churning, washing, and heating (Figures 2 and 3) [95]. After the ripe shea fruit is collected, it is de-pulped. The fresh nuts are then boiled to increase the fat output: this action softens the nut, disrupts cells, and promotes the better release of the oil [96]. Subsequently, the nuts are allowed to dry in sunlight (5–10 days) or in an oven (2–3 days) [24]. Deshelling involves removing the shells from the nut with a stone or hammer and, finally, removing the broken kernels and those infected by mold to obtain clean shea kernels [97].

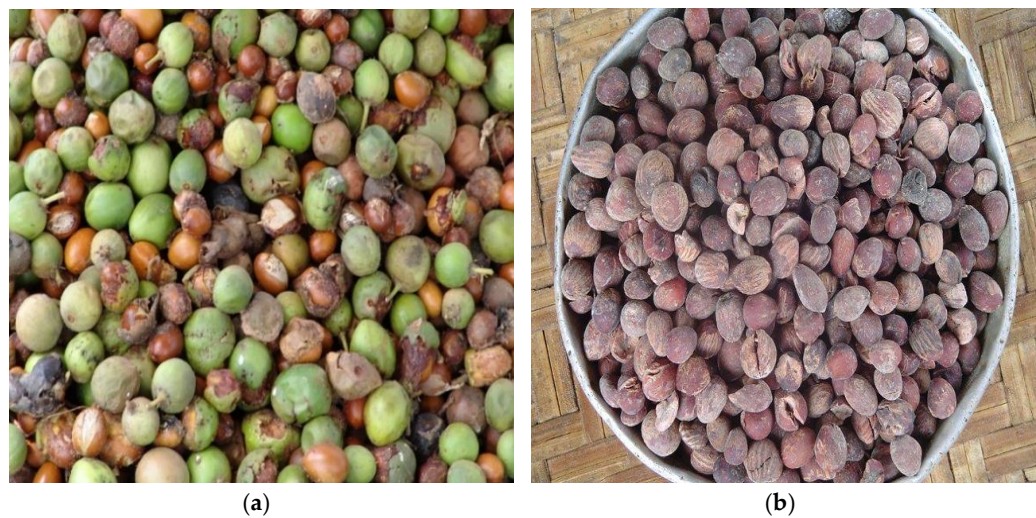

(**a**)  (**b**)

**Figure 2.** Vitellaria *paradoxa* fruit (**a**) and dry kernels (**b**). Photos: E.M.

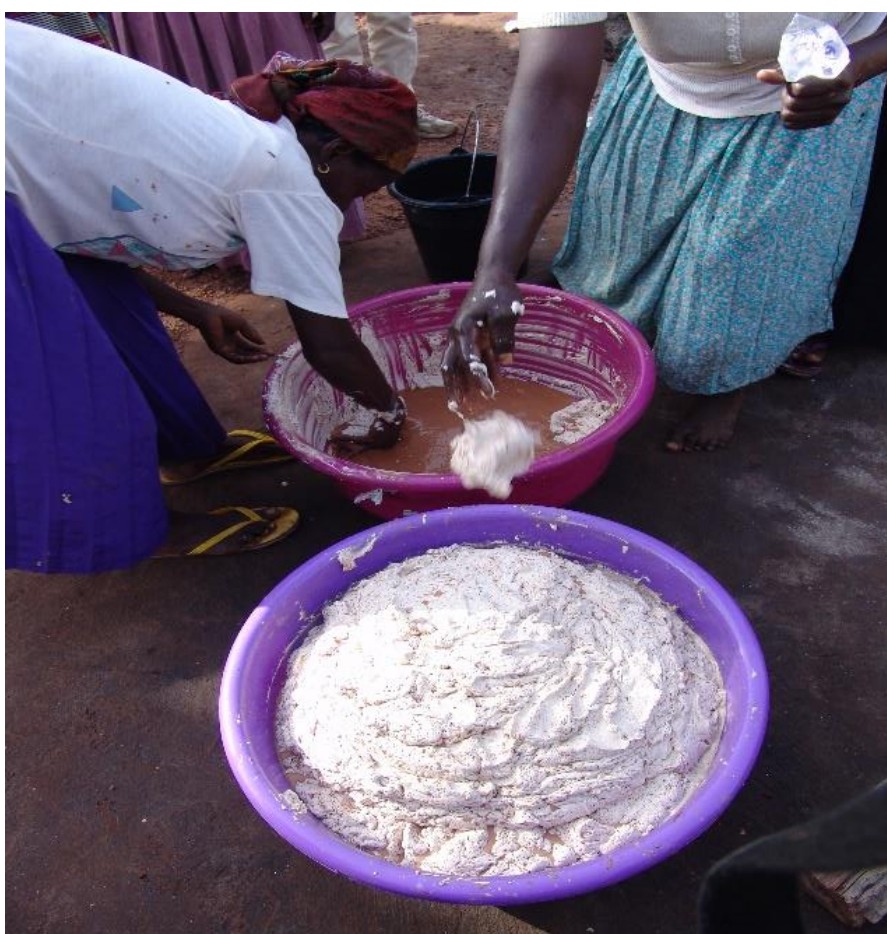

**Figure 3.** Shea butter extraction by kneading (*barattage*) in West Africa. Photo: E.M.

The extraction technique to obtain shea butter comprises three processes: kernel size reduction, kneading and heating, and, finally, oil separation and cooling [97]. Kernel size reduction involves pounding the kernels in a mortar using a pestle and further dehydration by roasting [98]. The roasted kernels are ground to a paste. Next, kneading involves adding a reasonable amount of shea paste to an initial amount of 3 L of cold water, stirring slowly, and then stirring vigorously by hand until the butter begins to rise in a crude, milky-white form. After kneading, the oily layer is harvested from the surface of the water layer. The oily layer is washed, boiled to evaporate the water, and the crude fat is obtained by decanting. The decanted oil is allowed to solidify—a process that takes 6–12 h—and the final product is shea butter [97,98]. Rural women in the household fully carry out the traditional processing of shea tree nuts. Shea butter extraction requires a lot of hard work, and due to its traditional processing, the quantity of butter obtained is variable, and yields vary considerably. As an approximation, 20 kg of fresh fruit may yield 4 kg of the dried kernel, from which 1 kg of butter can be extracted [95,99]. Figure 4 represents the flow chart of the shea nut fruit treatment into butter.

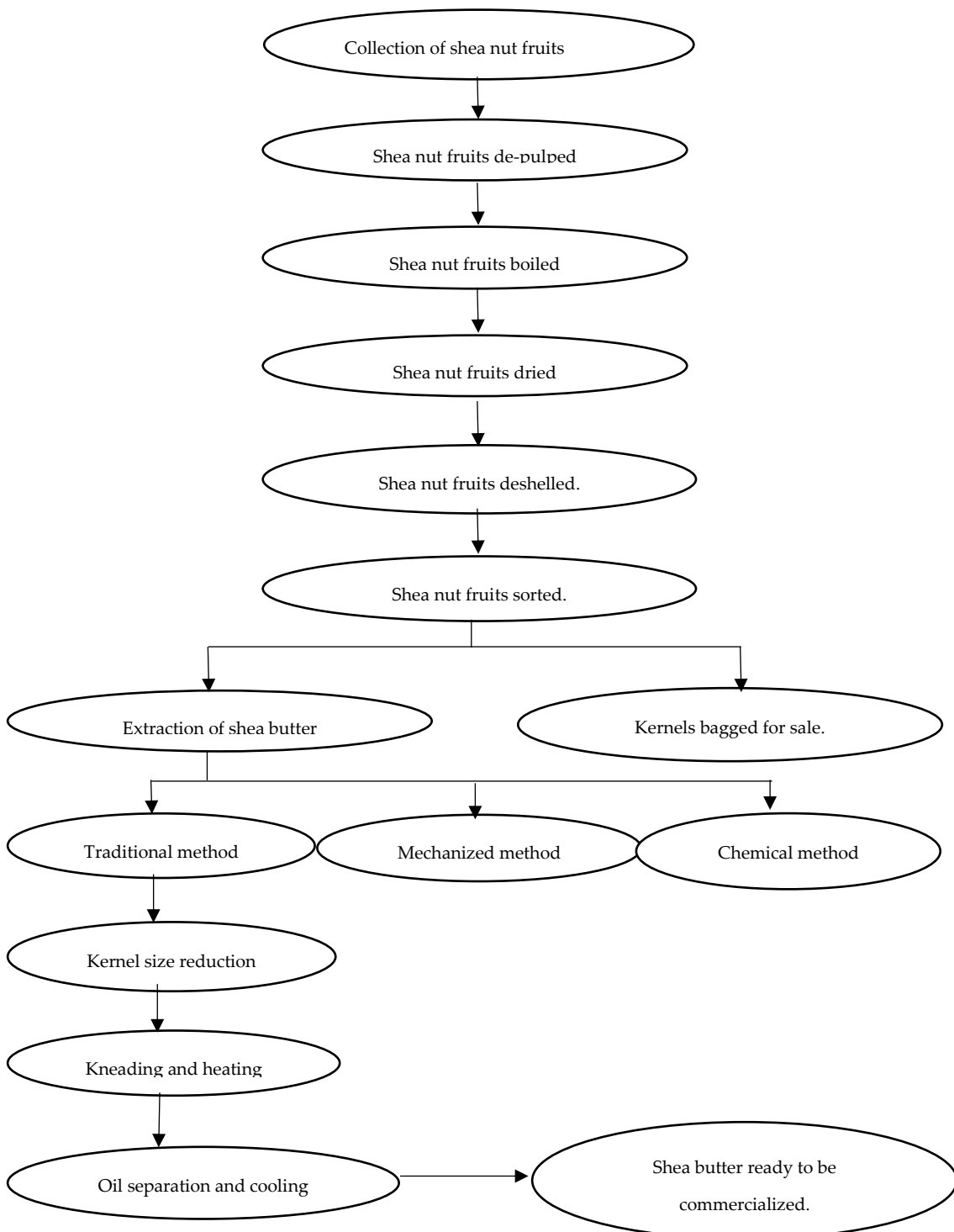

**Figure 4.** Shea nut fruit processing (adapted from [98]).

## 6. Nutritional Composition

### 6.1. Nutritional Composition of Fruit Pulp

Across its range, shea tree fruit pulp is of high dietary and nutritional importance [72,100]. Different authors have reported variable nutritional composition of shea fruit pulp with different methods used for estimation. This variability is perhaps due to agroecological differences within the populations, or genetic differences (Table 3).

Some researchers have estimated the moisture content of the dry shea fruit pulp to range from 8.8% to 72.4% [77,101]. A more significant moisture percentage of 67–80.3%

has also been recorded. The variation of moisture content can be attributed to processing, storage methods, and seasonality. The following figures regarding shea tree fruit pulp are compared with *Dacryodes edulis* fruit. The fruit pulp contains the following macromolecules: crude fibre, 3.06 ± 0.16 mg/100 g dry weight (dw) compared with 42.2 mg/100 g dw for *D. edulis*; crude protein, 19.23 ± 0.76 mg/100 g dw compared with 4.2 mg/100 g dw for *D. edulis*; and carbohydrates, 15.03 ± 0.45 mg/100 g dw compared with 8.1 mg/100 g dw for *D. edulis* [77,102]. Other compounds, including calcium and iron, can be found in the shea tree fruit [72,103]. Therefore, it is an essential local fruit to improve the livelihood of local populations [104].

**Table 3.** Nutritional and mineral composition of shea butter tree (*Vitellaria paradoxa*) raw pulp.

| Composition/Minerals | (Alu and Randa 2019) [a] | Muotono et al. (2017) [b] | Raimi et al. (2014) [c] | Aguzue et al. (2013) [d] | Okullo et al. (2010) [e] | Ugese et al. (2008) [f] |
|---|---|---|---|---|---|---|
| Moisture (%) | NE | 67 | 2.84 ± 0.13 | 4.58 | 72.4 ± 0.1 | 8.8 |
| Energy (kJ/100 g dw) | NE | NE | 2348± 0.41 | NE | NE | 198.8 |
| Carbohydrates (mg/100 g dw) | NE | 8.1 | 21.8 ±0.27 | 72.0 | 19.4 ± 0.6 | 42.5 |
| Crude protein (mg/100 g dw) | 15.2 ± 0.63 | 4.2 | 9.3 ± 0.05 | 3.5 | 3.1 ± 0.1 | 3.5 |
| Ash (mg/100 g dw) | 3.75 ± 1.10 | 5.1 | 4.18 ± 0.10 | 8.95 | 3.6 ± 0.2 | 4.6 |
| Crude fibre (mg/100 g dw) | 6.38 ± 0.10 | 42.2 | 12.68 ±0.13 | 9.6 | 14.5 ± 1.7 | 39 |
| Crude fat (mg/100 g dw) | NE | 1.3 | NE | NE | 1.5 ± 0.7 | NE |
| Ca | 0.18 ± 0.01 | 117.3 | 30.24 ±0.04 | 2.3 | 69.4 ± 0.1 | NE |
| Cu | NE | 0.1 | 0.80 ± 0.00 | NE | NE | NE |
| Fe | NE | 8.5 | 52.00 ±0.11 | 0.02 | 3.6 ± 0.1 | NE |
| K | 0.36 ± 0.02 | 830.3 | 61.70 ±0.30 | NE | 47.9 ± 0.2 | NE |
| Mg | 0.27 ± 0.02 | 57.2 | 6.24 ± 0.01 | 0.5 | 18.1 ± 0.3 | NE |
| Mn | NE | 0.6 | 0.30 ± 0.02 | 0.2 | NE | NE |
| P | 0.35 ± 0.02 | 39.8 | NE | NE | NE | NE |
| Na | 0.09 ± 0.01 | 19.3 | 5.10 ± 0.01 | NE | 8.9 ± 0.1 | NE |
| Zn | NE | 2.1 | 0.72 ± 0.00 | NE | NE | NE |

Note: All values are presented as the mean ± standard deviation. Abbreviations: dw—dry weight; NE—not examined. Source: [a] [73]; [b] [77]; [c] [105]; [d] [106]; [e] [103]; [f] [101].

A crude protein content has been reported in the range from 15.2 g/100 g dw [73] to 3.5 g/100 g dw [106]. A crude lipid content of 4.2 g/100 g dw and a crude fibre content of 42.2 g/100 g dw have been described [77,101,107]. An ash content has been reported from 4.7 to 5.4 g/100 g dw [79,101]. The information presented could help to have an overview of the vital energy and nutritional source derived from the shea fruit pulp.

### 6.2. Biochemical and Phytochemical Composition of Shea Kernels and Butter

The average moisture content of dried shea kernels has been reported as 6.8% and 1.4% for the kernel and butter, respectively (Table 4). Kernels contain nearly 50% of fats that, after processing to butter, rise to 75%. It has a moderate mineral content of magnesium and calcium both in kernels and butter. The whole kernels are not edible due to their high levels of antinutritional factors, including tannins and catechins.

**Table 4.** Composition of shea kernels and butter. Adapted from Honfo (2015).

|  | Kernel | References | Butter | References |
|---|---|---|---|---|
| **Macronutrients** |  |  |  |  |
| Moisture (%) | 6.8 | [108–110] | 1.4 | [75,111–114] |
| Carbohydrates (g/100 g dw) | 30.9 | [109,111,115] | 22.3 | [113] |
| Crude protein (g/100 g dw) | 8.1 | [109,111,115] |  |  |
| Crude lipids (g/100 g dw) | 45.2 | [100,110,116,117] | 75 | [113] |
| Crude fibre (g/100 g dw) | 9.1 | [111,115] |  |  |
| Ash (g/100 g dw) | 2.5 | [111,115,118] | 2.3 | [113] |
| **Minerals** (mg/100 g dw) |  |  |  |  |
| Ca | 71.8 | [109,112,115,119] | 9.6 | [112] |
| Cu | 0.3 | [112] | 0.8 | [112] |
| Fe | 1.6 | [112,115] | 3.6 | [112] |
| K | 0.1 | [119] | 2.2 | [112] |
| Mg | 142.6 | [112] | 4.5 | [112] |
| Mn | 0.4 | [119] | 0.006 | [119] |
| P | 0.04 | [109,115] |  |  |
| Zn | 0.9 | [112] | 2.7 | [112] |

Abbreviation: dw—dry weight.

The components of shea butter that affect its physicochemical properties are triglycerides and a large fraction of unsaponifiable compounds that are recognized as active principles in cosmetic products [120]. An unsaponifiable compound is a fraction identified in shea butter that does not dissolve in acetone. It could also be designated as a highly unsaturated compound that consists of a mixture of different polyisoprenes [121,122]. The average unsaponifiable content differs, depending on the authors, from 1.2% to 17.6% [112,123]. The acid value in the butter determines the way glycerides are decomposed by lipase; therefore, it assesses the impact that heat and light cause on the components. The results also help to indicate the oil quality. The value varies with an average of 8.1 mg KOH/g [124]. The free fatty acid (FFA) percentage has also been evaluated, ranging from 1% to 10.7% [125].

Authors have surveyed the fatty acid content of the butter. It is characterized by a total of 16 saturated and unsaturated fatty acids, with 5 of them (oleic, stearic, palmitic, linoleic, and arachidic) showing relatively high levels [100]. Oleic acid is dominant, from 37.2% to 60.7% [117,126], followed by stearic acid, varying from 29.5% to 55.7% [75,103,117]. The quantity and quality of fatty acids differ from one region to another. Authors have revealed that oleic acid is more dominant in butter coming from Uganda, while stearic acid is more dominant in butter coming from West African provenances [51,95,100]. Palmitic acid varies from 3.3% [100] to 7.5% [103]. The linoleic acid content is between 4.3% and 8.0%. This fatty acid is essential in nutrition because it is a component of the cell membrane [110,118,127]. Arachidic acid varies between 0.8% and 1.8% [51,100,103]. The data provided reveal a significant variation in the composition of kernels and butter.

An assessment of the phytochemical diversity and quality of shea kernels has revealed that they are rich in anti-inflammatory phytochemical constituents [88]. Triterpenoids in kernels (α-amyrin and β-amyrin) have anti-inflammatory activities [128]. Terpenoids are credited as having the inhibitory potential of nuclear factor kappa-light-chain-enhancer of activated B cells (NF-κB) to reduce inflammation and counteract cancer [129]. Shea kernels contain triterpene acid, triterpene glycoside, steroid glucosides, and other phenolic compounds, as active principles. There are also positive aspects of extracting specific compounds from defatted shea kernels that could be used as antioxidants and anti-inflammatory agents [35]. An extract of the nutshells has been investigated as a potential

source of medicines for treating diabetes, based on the presence of bioactive compounds including protocatechuic acid, trihydroxycoumarin, taxifolin, and quercetin. These chemicals have been investigated via the molecular docking method for high-performance liquid chromatography (HPLC) fingerprints and bioactivity evaluation [130]. According to the authors, nutshells could provide a source of cheap, natural antidiabetic compounds and are especially rich in (2*R*, 3*R*)-(+)-taxifolin, which would help to reduce waste of the currently discarded shells and increase the profitability of tree growing.

The fatty acid composition of triterpene alcohol fractions of the non-saponifiable lipids of shea tree kernels has also been studied. The dominant fatty acids are stearic (28–56%) and oleic (34–61%) acids. Triterpene contains a range of components: α-amyrin, β-amyrin, lupeol, and butyrospermol as the major constituents [35,117,120,131,132].

Secondary metabolites are present in the shea tree's plant parts (leaves, bark, stems, roots, and fruits) [133]. The phytochemical analysis of leaves and bark extract of shea tree have revealed compounds, such as alkaloids, flavonoids, tannins, steroids, phenols, phlobatannins, glycosides, and carbohydrates [76,116,134–136].

## 7. Importance of *V. paradoxa* in the Context of Agroforestry

The shea tree is considered one of the principal tree species primarily found in agroforestry parklands in sub-Saharan African countries, where the species occurs naturally [137]. It constitutes a high percentage of standing biomass, contributes to reducing soil degradation, and possesses a significant carbon sequestration ability, which can be used for climate change mitigation strategies [59,138–140]. The shea tree is the most common in Mali and has a relatively high density of 30–50 trees/ha [53].

Growing shea trees in agroforestry plantations impact the microclimate positively. Due to trees shading effects, there is an essential effect on the soil moisture content, leading to higher crop yields [87]. Some researchers have observed higher fruit yields in cultivated fields with shea trees compared with naturally fallow land. Cultivated fields with annual crops growing where shea trees were present provided better flowering conditions, fruiting, and better yields [3,141,142]. In Burkina Faso, 51% of agricultural land was identified as suitable for growing the species [143]. The cultivation of crops such as maize and soybean is being used to generate income and to enhance food security in shea tree parklands. The authors of a study carried out in Uganda suggest that different responses have been observed regarding soybean and maize yields [144]. Due to the competition for light and nutrients, intercropping with mature shea trees led to very low maize and soybean yields compared with intercropping with young shea trees. Another investigation has been carried out regarding the influence of the shea tree and *P. biglobosa* trees on sorghum production in Burkina Faso [144]. The authors found that sorghum yield was affected negatively by 50–70% with *P. biglobosa*. According to some studies, as shea trees age, there is a reduction in the yield of adjacent crops. The yield variability could be explained by the shading and competition for water and nutrients, but the cultivation of soybean in shea tree parklands is a possible recommendation as lower reduction in yields were reported compared with cereals [144–146].

## 8. Silvicultural Management of *V. paradoxa*

### 8.1. Tree Management

Local farmers selectively conserve valuable trees on the farm when clearing the vegetation to prepare their fields [99]. This practice has created a typical landscape dominated by shea trees and other associated trees such as *Acacia* spp., *Faidherbia albida*, and *P. biglobosa* [82]. The selection of shea trees during field clearing is based on complex criteria that include spacing, phytosanitary conditions, fruit characteristics, and yield characteristics for each individual [147].

The fructification status of some populations of *V. paradoxa* trees has been assessed in Burkina Faso [148]. The authors revealed that 94% of trees grown in agroforestry parklands produced more fruits than natural stands. The average yield of kernels per tree was

significantly higher in agroforestry parklands than those from trees in natural stands. The yield appeared to be higher, probably because of the presence of other trees species on the same agroforestry plot. In another study, the authors evaluated the distribution and spatial pattern of shea trees based on farmers' practices in Mali [149]. They compared three types of land use systems (forest, cultivated field, and land that had been fallow for 4–5 years) where shea trees occur. They found significant variation in tree diameter among land-use types. Shea trees found in the forests had a diameter <50 cm. In cultivated fields, they had a diameter of 80–130 cm, and on fallow land, they had a diameter of 50–150 cm. The differences are probably caused by a high level of competition for light and nutrients between shea trees and other tree species. Therefore, that competition is higher in the forest than in the other systems. The management of shea trees in cultivated fields or recently fallow land requires innovative approaches. Farmers have to identify young shea tree plants that have grown naturally or suckers (young shea twigs from a dead stump), till in the soil around in the form of a bowl, and put protective structures around the young plants [150].

*8.2. Propagation and Growth Performance*

8.2.1. Generative Propagation

Generative propagation is the oldest propagation technique; it is commonly performed by directly growing seeds in nurseries. The individuals obtained from this technique do not look like the mother tree [151]. The germination of shea tree nuts is relatively low, and the seeds have poor longevity [152].

The authors in Bayala et al [153] examined the growth variation in shea tree seedlings under imposed drought stress in Burkina Faso. They aimed to assess the ability of juvenile shea tree plants under severe drought stress. The parameters tested were the size and origin of the seed, seed germination, and survival rate. According to sampling locations, there were some differences in the plants' morphological variables (leaves and above ground parts). Water stress negatively affected the aerial growth parameters such as the height and diameter of the plants. The authors observed no correlation between seed origin, seed characteristics, seed germination, and the survival rate of seedlings. Parameters such as root length and seedling traits are essential factors in the initial growth of shea tree seedlings. Other researchers studied the growth performance of shea trees on seedlings with supplementation of mineral nitrogen, phosphorus, and potassium (NPK) fertilizers and arbuscular mycorrhizal fungi. The effects of NPK fertilizers had a relatively low impact on the shea tree root and growth ability. According to the different treatments, mycorrhizal root colonization appeared to be low at ≤12% [154].

The effects of substrate and shelter type on the germination rate of shea tree seeds have been assessed. The results demonstrated that seeds could germinate in a greenhouse compared with shelters covered with straw. In the study, several parameters—different soil substrates, germination parameters (germination capacity, germination percentage, latency), and land-use types—were employed. To obtain a successful germination rate, the seeds must be in good health. Environmental factors such as temperature and humidity also need to be considered because they contribute to the activation of the hormones and enzymes required for germination purposes [155]. Shoots and roots developed through bipolar embryos were acclimated during in vitro culture in a greenhouse. The experiment aimed to assess the role of 2,4-D in inducing embryogenic callus in the shea tree, albeit at very low concentrations (0.45 μM). Unfortunately, the germination success rate was low (15%). This study provides results on developing a protocol for somatic embryogenesis and plant regeneration of shea trees and insights for further work on optimizing culture conditions in controlled areas [156].

8.2.2. Vegetative Propagation

The positive aspect of vegetative propagation techniques is a reduction in time to maturity, increased yields, and possible fruit production during the off-season [13,15,157,158].

Using improved material and vegetative propagation techniques, plant material can develop deeper soil roots to capture and absorb more water [71,154]. Techniques such as cutting, grafting, air layering, and in vitro propagation have been evaluated.

- Stem cuttings:

The non-mist propagator is one of the most suitable pieces of equipment for testing the rooting ability of stem cuttings. It is well designed and adapted to propagate a structure because it could give a successful (70–85%) rooting percentage [157,159–162]. In Amissah et al [163], the authors revealed that rooting ability depended on the maturity level of the cutting source. Cuttings from mature trees showed poor rooting ability (27.9%) compared with 45-year-old trees (40.9%) [163]. Another study evaluated the effect of rooting media and different concentrations of indole 3-butyric acid (IBA) on root formation in cuttings of shea trees [164]. IBA concentrations had a strong effect on the rooting ability of shea tree cuttings. The authors reported that 30,000 ppm gave better rooting (57.5%) than 7000 ppm (30%) and the control without IBA (7.5%). Yeboah et al. [165] conducted a study at the Cocoa Research Institute of Ghana about the rooting performance of shea trees using different rooting substrates and hormones. They revealed that rooting ability was significantly higher using rice husk as a rooting substrate compared to substrates such as sand–sandy loam (1:1). The rejuvenated shoots that were not dipped in water gave a significantly higher rooting percentage than shoots not submerged in water. On the other hand, the use of indole-3-butyric acid (IBA) powder provided more and longer roots per cutting than the control ($p < 0.05$).

- Grafting:

The grafting technique has provided a high success rate for shea trees in various studies. The results are drawn from experiments with five methods of grafting used in farmers' fields. The reported success rates are: 86.1% (side cleft), 80.9% (side tongue), 78.1% (top cleft), 38.1% (chip budding), and 20.7% (side veneer) in Mali and Burkina Faso [166,167].

- Air layering:

Air layering could be defined as the rooting ability of the branch by peeling the bark and applying a mixture of soil and sand, and then covering it with a transparent paper for the development of the roots. In experiments conducted at the Cocoa Research Institute in Ghana, researchers have demonstrated a success rate of 33.3% and 22.2% with softwood and semi-hardwood, respectively [25,168]. However, this technique cannot be implemented on a large scale due to the relatively lower success rates.

- In vitro propagation:

Shea tree explants cultured on shoot regeneration media containing Murashige and Skoog (MS) macro- and micro-nutrients at full- or half-strength showed no difference in the number of axillary shoots induced. The roots were influenced when regenerated axillary nodes of the shea tree. They were cultured for 70 days on root regeneration media containing MS macro- and micro-nutrients at quarter strength. No signs of rooting were observed when shoots were cultured on media containing MS macro- and micro-nutrients at half strength. While the success rate was low, it should be noted that 76% of cultures were still healthy upon completion of the experiment. On the other hand, the adventitious shoot formation in vitro, originating from the apical shoots of seedlings, is a valuable means of multiplication because young trees of this species usually exhibit only apical growth axillary branches forming after 5–6 years [169].

- Plan for rapid multiplication and how to shorten the long cycle:

The shea tree is one of the priority key species under the tree domestication program by World Agroforestry. There are ongoing experiments regarding the multiplication of the species.

The vegetative propagation technics include the rooting of cuttings, air-layering, and grafting. They are proven records that these methods allow to produce individuals with the same genetic characters with desirable genotypes [165]. While using these methods, the fruiting of the shea tree was observed after 3–4 years with young cuttings and grafted seedlings. From the observations and literature reviews, vegetative propagation techniques can therefore be the ideal option to enhance the domestication and rapid multiplication of the shea tree.

*8.3. Silvicultural Management of Trees*

Farmers carry out the establishment and management of shea agroforestry parkland. The establishment occurs when the field is cleared, and the valuable trees are selected purposefully and protected from the natural vegetation. The shea tree has a long juvenile phase lasting from 10 to 25 years. The cultivation of this species by seedlings is slow and difficult because it will reach maturity after 30 years. The life span of the tree is around 250 years [25].

Regarding the management of vegetation, most farming activities contribute to the structure of agroforestry parkland to enhance the flowering of trees. In Burkina Faso, fruit yield and measurements were analyzed during the year. Yields were higher in agroforestry parklands (4.3 kg/tree) compared with the other natural woodlands (1.6 kg/tree). The average tree diameter, fruit size, and weight were more considerable in parklands than in natural plots. These outcomes are probably due to a combination of factors—selection of the best trees, cultivation practices promoting soil nutrient improvement, the reduction in tree competition through thinning, and clearance of vegetation and tree pruning [148]. Some pests and diseases always threaten the production (yields) of shea fruit.

8.3.1. Parasitism

The main common infestations are with the hemiparasitic plants (Loranthaceae), associated with three species of mistletoe parasites: *Agelanthus dodoneifolius*, *Tapinanthus globiferus*, and *Tapinanthus ophiodes*. High infestation rates have been reported in the West African parklands of Burkina Faso (95%) [170] and Nigeria (81%) [171]. In northern Cote d'Ivoire, two parasitic species of Loranthaceae have been identified—*Tapinanthus bangwensis* (Engl. and Krause) *Danser,* and *A. dodoneifolius* (DC.) Polh et Wiens—with infestation rates from 59.7% to 65.5% [172].

8.3.2. Diseases and Pests

The shea tree has low resistance to diseases and pests. The causative organisms mostly attack the leaves, where they cause dark brown and grey spots. The identified pathogens are: *Fusicladium butyrospermi* Griff. and Maubl and *Pestalozzia heterospora* Griff. and Maubl [25,173]. From pests, *Salebria* spp. (Lepidoptera: Pyralidae) have also been identified, with infection rates of 49–80% for trees and 4–15% for fruit in Burkina Faso [174].

*8.4. Improvement Plan of V. paradoxa*

Concerning the improvement plan and conservation strategies, the following specific objective will include:

- Strengthening forest management institutions through a local participatory approach for a suitable management plan about essential non-timber forests products (NTFPs) with high potential in the region (e.g., shea tree).
- Drawing a regional map of the model for sustainable management of NTFPs with a focus on livelihood improvement.

Some of the local strategies to conserve shea trees on the farms are the following: (i) raising them deliberately of the farms; (ii) allowing natural regeneration to grow on the farms; (iii) distracting local farmers attention to cut down the shea trees; (iv) protecting shea trees from pests and diseases using local strategies; (v) staking young shea trees from destruction by livestock; (vi) setting up boundaries of gardens with shea trees.

## 9. The Role of *V. paradoxa* in Food Security and Socio-Economic Development Diseases and Pests

The shea tree is recognized as a species that contributes to the economic growth and socio-economic development in SSA countries [24]. As already mentioned, the shea tree is one of the most valuable fruit tree species utilized in agroforestry parkland across SSA [56,175]. Many parts of SSA are suffering from hidden hunger, malnutrition, and food insecurity [176]. From the fruit harvested from shea trees, 25% is used as a local source of food, and the remaining 75% is sold for cash in the local markets [79,177,178].

Concerning shea collection, women are more involved than men. It is estimated that as many as 16 million women, half of them in West Africa, are involved in shea-related activities [65]. The report of the Global Shea Alliance (GSA, 2020) provides data showing that 4 million women are involved in the export value chain, with 200 million USD generated as income every year in communities that produce shea tree products. According to the data provided through the same report, the GSA explains that women collectors collect an average of 4 bags each with 85 kg of kernels per season, with 2 bags traded as kernels to intermediaries and 2 bags used for local processing of butter for local markets [65,179]. Therefore, the nuts are stored for later processing in dry areas (tiny houses constructed as storage) after the women have collected them from the field. Annual shea kernel production in Burkina Faso ranges from 70,000 to 300,000 tons [94,180].

Both women and men have commercialized the shea tree [181,182]. The commercialization of shea tree kernels contributes 66% to household income in Burkina Faso [178]. Moreover, from the kernels collected, 41–55% are consumed by domestic households and therefore not included in trade statistics [64]. In Côte d'Ivoire, the purchase price per kilogram of shea kernels varies from approximately 0.05 to 0.10 USD between the season's start and end (June–October). The estimated gain achieved by a butter producer during the shea season is between 156 and 183 USD. Prices noted by urban market traders indicate that the selling price of shea butter from the processors at the start of the season is 0.18 USD, and at the end of the season, it is 0.74 USD. From wholesalers, the price varies between 0.74 and 1.10 USD. During the shea season, the average estimated profit of shea kernels and butter trader is 84.32 USD per month. Likewise, the wholesalers of shea products have reported that they can make an average profit between 7332 and 18,330 USD per season [183]. Women use the income generated from the commercialization of shea tree nuts to meet the family's daily needs, such as food, their children's education, and health care [178]. The commercialization of shea tree nuts by women could also ensure rural development in the community [25].

Shea tree cultivation and management appear to be influenced significantly by gender, family size, and farm size. According to Okiror et al. [184] in Uganda, the bigger the family size (6–9 members), the more resources are given towards shea management. The bigger the farm size (at least 9 ha), the more willingness to manage the species because the household size determines the ability to satisfy basic needs. Therefore, the household's key decision makers in shea tree management are men (55%) and women (35%); other family members account for the remaining 10%. This imbalance is because men are regarded as owners of the family's land and the most influential; hence, in most cases, men have the discretion to plant trees or cut them down.

The shea tree is considered one of the most important tree species in dryland areas. The species significantly improve the livelihood of local smallholder farmers through its derived products, with shea butter as the main product. The pulp of the fruit is highly nutritious and is recommended in daily diets. The income generated from selling shea tree products helps reduce poverty, while other cash crops increase food security.

## 10. Morphological and Genetic Diversity

Genetic diversity is of great importance in tree domestication. It is used as a resource in tree breeding: to improve local tree species and contribute to the genetic gain for many traits. It also contributes to the livelihood improvement of local communities using

traditional knowledge and provides insights regarding sustainable development [185]. Hence, scientists have used phenotypic studies and environmental characterization of genetic variation to assess fruit properties [2,27–30].

In a study carried out in Burkina Faso, the authors examined the value of gender-sensitive participatory research to understand local botanical knowledge and preferences of shea trees [38]. A total of 25 ethno-variety trees were identified, based on 11 types of fruit using morphological descriptors. The parameters are the pulp (taste, color, texture, and quality), fruit characteristics (size and shape), and nut traits (size, number of seeds, color, and quality). Among all preference ranking groups for ethno-varieties, 'big shea fruit' appeared to be most preferred by the farmers. According to the gender perspective, men preferred 'big' and 'long' shea tree fruit compared with women, who preferred 'big' shea tree fruit. One of the first studies was carried out in Cameroon, focusing on the phenotypic diversity of the shea tree [186]. Considering the quantitative characters, populations of the Northern regions (low altitude) had significantly larger circumferences of the trunk and significantly smaller diameters of the fruit and nuts than those of the Adamawa and West regions (high altitude). The diameter of the fruit varied from 37.33 to 82.47 mm, and the nut was 29.57 to 53.34 mm from low altitude to high altitude. The assessment of morphological diversity among their population showed a significant correlation between traits according to agroecological locations.

Further results demonstrated a 77% chance of having larger fruit at a high altitude and a small trunk circumference. Their large trunk and smaller fruit recognized trees present in low altitudes. The frequency of the different phenotypes varied from one site to another. Four tree shapes—broom-shape, ball-shape, broadly pyramidal shape, and oblong shape—were identified; most of the trees were ball-shaped with ovoid fruit [183]. In Côte d'Ivoire, the trees were mostly broom-shaped with ovoid fruit [187].

Authors have also assessed the morphological and genetic diversity of shea trees in Ghana using several phenotypic traits and ten microsatellite markers [188]. The genetic data revealed high genetic diversity based on mean expected heterozygosity and allelic richness, measured as several effective alleles. The land-use systems (farmland, forest) studied here were ranked as very diverse. The authors reported that farmland trees were significantly bigger than trees from natural forest trees. This difference could be because the farmlands contained bigger and older trees in the agroecological area. This diversity indicates the lack of negative influence of farmers' selection on genetic diversity. There were some differences in morphological traits between land use and agroecological zones [188]. In another study, the authors investigated genetic variation between and within populations using microsatellite markers and quantitative traits of the shea tree in Mali [189]. The main findings revealed a low genetic diversity ($H_e$ = 0.25–0.42) compared with other tree species, such as *Grevillea macleayana* ($H_e$ = 0.42–0.53) and *Symphonia globulifera* ($H_e$ = 0.83), assessed with the same markers. The status of being a socially and economically important species could cause this lower genetic diversity. This low genetic diversity was caused by human activities and management practices carried out in the area over the past years. Another reason is the protection (from fire and competition) granted to shea trees during long-term management. The fruiting, flowering, and density of shea trees are more effective in parkland systems than natural stands.

In the same area, other researchers evaluated the variation of agro-morphological traits of the shea tree [53]. Leaf and fruit qualitative characteristics were positively correlated with the amount of rainfall. The correlation between fruit pulp and nut weight was low compared with other fruit traits. The different provenances could explain the reason for this phenomenon. The results therefore indicate a very low genetic variation between the group of traits. There was also a correlation between the circumference (size) of shea trees and annual rainfall: shea trees were bigger in the dry zones. The genetic correlation was not well estimated in this study because the genetic and environmental parameters were not separated. In a survey undertaken in Benin, the authors reported a significant difference in shea trees in different land-use types (park, hunting zone, and farmlands) [190]. The shea

trees from farmland had the highest production of fruit. This outcome occurs because trees in farmlands could express their full potential, a factor that ensures good production. In contrast, low production in parks could be caused by burning for management purposes, competition with other plants, and damage from various wild animals. There was high variability between leaf sizes of trees within the sites, while for the fruit, the most significant variability was among fruit from the same tree.

The shea tree has been characterized in Mali using microsatellite markers. The observed level of heterozygosity has ranged from 0.04 to 0.51 [191] and 0.37 to 0.85 [192]. These results specify a moderate genetic diversity among the populations sampled. Another study in Uganda [193] also demonstrated the level of heterozygosity, with values of 0.633 and 0.727, demonstrating higher genetic variation. Additional results considering the analysis of molecular variance showed that most of the variation occurs within the individual trees (86.28%).

Moreover, 11.25% of the variation is observed among individual trees within ethno-varieties. The remaining 2.4% of the variation is found among the ethno-varieties. Molecular data have revealed isolation by distance and past population expansion for the shea tree. Experiments have been carried out on samples collected around seven West African countries: Benin, Burkina Faso, Ghana, Côte d'Ivoire, Mali, Nigeria, and Togo [194]. The diversity found was moderate to high within populations. Another result was the weak differentiation among populations. Based on allele frequencies, the neighbor-joining tree confirmed a low differentiation among populations with no phylogeographic signal detected with nuclear microsatellites.

Further sequencing and transcriptomics demonstrated high lipid content in shea tree fruit [195]. This phenomenon could be explained through the genome of the shea tree, which could encode more lipid biosynthetic genes, such as ketoacyl-ACP synthase genes. The genes found in shea tree fruit through biosynthesis transcriptomics and functional heterologous expression are triacylglycerol (TAG) and 1,3-distearoyl-2-oleoyl-glycerol (SOS, C18:0–C18:1–C18:0).

RAPDs and chloroplast microsatellites have been used to quantify the genetic variation of shea trees sampled in the natural range (Benin, Burkina Faso, Côte d'Ivoire, Mali, Senegal, Central Africa, Cameroon, and Uganda) [196]. The results demonstrated no relationship between diversity and longitude or latitude. A Mantel test revealed a positive correlation between genetic distances and geographic distances ($R = 0.88$, $p = 0.001$). In another study, three chloroplast microsatellite primers, assayed in 116 individuals, revealed 10 different alleles and 7 chlorotypes among the different populations. The phylogeography of the shea tree has been assessed through past climate changes over Africa [197]. The findings revealed that haplotypic and allelic richness could vary significantly due to chloroplast and nuclear microsatellites, which point to a higher diversity in West Africa. There is an equally important level of differentiation among the populations. The climate variations are the major factor explaining the genetic pattern of the shea tree.

## 11. Conclusions

This review paper summarizes the available information about the current state of knowledge about *V. paradoxa* research. It enhances our understanding of how existing methods for sustainable management practices have improved and how resources, especially kernels, could be transformed into valuable products, including comestible butter. The obtained shea butter is therefore widely commercialized worldwide. The species is used in various domains and provides valuable services for both livelihood improvement and environmental benefits. The species is beneficial and has proved to be crucial in future research in SSA. The intense production of high-quality shea butter could encourage the consumption of other wild fruits and NTFPs that are underutilized, contributing significantly to the daily life of local communities. As *V. paradoxa* fruit pulp is a good source of nutrients and phytochemicals, it is highly recommended to conserve the genome of this species. The species is also on the red list of threatened species to be vulnerable,



according to the IUCN. Additional studies are recommended to develop tree improvement and conservation programs. The next gaps that need to be filled include the physiology and reproductive biology of the shea tree; the role of the shea tree in household income, food security, and women's empowerment—none of which have been explored in detail; the contribution of ethnobotany in biological conservation; and genetic studies using recent genomic tools and applications for breeding purposes. Concerning the ongoing domestication programs, it is crucial to connect genetic studies with the AOCC to contribute to the sustainable management of shea tree populations across the West Central African belt. The Czech University of Life Sciences (CZU) in Prague is currently carrying out an important project collaborating with scientists from World Agroforestry in Kenya and the Higher Institute of Environmental Sciences (HIES/IBAYSUP) in Cameroon. This research aims to sequence the shea tree genome using recent technologies where biparental populations will be assembled. The data obtained will generate high DArT-seq/single nucleotide polymorphism markers for linkage mapping among populations in Cameroon and Mali. Hopefully, this endeavor should contribute to the sustainable management of the existing germplasm, leading to the greater economic security of vulnerable smallholder farmers. This review has identified gaps that should be the subject of future research about the shea tree while considering different provenances.

**Author Contributions:** Conceptualization, P.B.C.N. and B.L.; Methodology, P.B.C.N., P.H., Z.T., M.K., A.V.T.T., D.K., E.M. and B.L.; Validation, P.B.C.N., E.M. and B.L.; Investigation, P.B.C.N.; Data Curation, P.B.C.N., P.H., E.M. and B.L.; Writing—Original Draft Preparation, P.B.C.N.; Writing—Review & Editing, P.B.C.N., P.H., Z.T., M.K., A.V.T.T., D.K., E.M. and B.L.; Supervision, P.H., Z.T., E.M. and B.L.; Project Administration, P.H. and B.L.; Funding Acquisition, P.H. and B.L. All authors have read and agreed to the published version of the manuscript.

**Funding:** This research was funded by the Internal Grant Agency of the Czech University of Life Sciences, Prague (grant number 20213110); a scholarship from the Czech government; and the research fund provided by World Agroforestry.

**Acknowledgments:** Special acknowledgment to Tsobeng Alain from World Agroforestry (ICRAF), Yaounde-Cameroon office, and Maňourová Anna from the Czech University of Life Science (CZU) for their help in improving the earlier versions of the manuscript.

**Conflicts of Interest:** The authors declare no conflict of interest.

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
