# Peer review of "The Current State of Knowledge of Shea Butter Tree (Vitellaria paradoxa C.F.Gaertner.) for Nutritional Value and Tree Improvement in West and Central Africa"

_forests, doi:10.3390/f12121740_

Round 1
Reviewer 1 Report
Please see the comments attached

Author Response
Dear reviewer,
Thank you for your feedback and comments.
Improvements were done according to the remarks, errors in punctuation and grammar were checked.
The manuscript will be uploaded for appreciation.
Reviewer 2 Report
The revised manuscript is much improved
Author Response
Dear reviewer,
Thank you for your feedback and comments.
The English language and spell check were done.
The manuscript will be uploaded for appreciation.
Reviewer 3 Report
The review paper presented here is of good importance for the community and has indeed summary different activities carried out on the pant.
However, I will recommend the authors to reformat the topic to make it broader. Also, in the abstract section, the authors emphasized on the genetic diversity and population structure but nothing much was presented in the review except section 9 (not much in case there is limitation information author should mention it as well). Some parts of the document are not necessary as most reflect to be common information. Section 10 should come before 9.
For a review, such information is missing and I would recommend authors to provide these basics information's in the document
- improvement plan and conservation programmes across the proposed study area, in-situ, ex-situ...etc.
- Physiology and reproductive biology of the shea tree
- the role of the shea tree in household income, food security, and women's empowerment
- Plan for rapid multiplication and how to shorten the long cycle
Round 2
Reviewer 3 Report
Authors have addressed well my comments and the manuscript is in good shape now.This manuscript is a resubmission of an earlier submission. The following is a list of the peer review reports and author responses from that submission.
Round 1
Reviewer 1 Report
This paper seems comprehensive and useful, although the section on genetic diversity is short and limited, leading one to question why genetic diversity appears in the title.
Overall, the paper needs to be heavily edited/rewritten by someone who knows the English language well and is experienced as an editor. In its current state, the paper is difficult to follow and understand. There is widespread redundancy, pieces of information being included multiple times, often in the same paragraph; sometimes, one statement contradicts the next. Paragraphs are jumbled; sentences do not necessarily follow in a logical sequence. Sentences are not grammatically correct and word choice is poor or inaccurate.
Reviewer 2 Report
Review manuscript: forests-1258130
This manuscript is a good monograph on an important nontimber forest tree species. It provides fundamental information about the species. People interested in knowing more about taxonomy, botany, ecology, phenology and uses would benefit from reading the manuscript.
The first part of the manuscript is well written with few grammatical challenges. The latter sections are less so. The paper reads as though different people contributed to the sections and it was not edited by one person. This needs to be addressed. Someone needs to go through the entire document and edit it appropriately to address the grammar and syntax issues.
Unfortunately, I feel that the manuscript is not appropriate for the journal. It does not address a particular research question or test an hypothesis. There are no methods, and the way the paper is presented none are needed. After reading the manuscript, the reviewer is left with the perception; good information and now what? Although it adds to the body of knowledge on the species, it does not address a problem or an issue.It does not critique the literature, or identify gaps in knowledge that require more research.
The manuscript is worth publishing, just not in this type of journal. I encourage the authors to pursue publishing through the African Orphan Crop Consortium (AOCC) or the World Agroforestry Center. These organizations have a vested interested in the information and should be able to produce a quality monograph. Both organizations are identified in the manuscript as committed to ‘the improvement of underutilized crops which have hitherto received little attention from the scientific community.” Publishing this manuscript as a monograph would demonstrate the organizations’ commitment to this goal.
Reviewer 3 Report
see attached
